# Drying Applications during Value-Added Sustainable Processing for Selected Mass-Produced Food Coproducts

**Huaiwen Yang** [1,*,†]**, Tulakorn Sombatngamwilai** [1]**, Wen-Yao Yu** [2] **and Meng-I Kuo** [3,*,†]

1   Department of Food Science, National Chiayi University, Chiayi City 60004, Taiwan; tulakorn.st@gmail.com
2   Texture Maker Enterprise Co., New Taipei City 251, Taiwan; bearep308@gmail.com
3   Department of Food Science, Fu Jen Catholic University, New Taipei City 24205, Taiwan
*   Correspondence: calyang@g.ncyu.edu.tw (H.Y.); 062998@mail.fju.edu.tw (M.-I.K.)
†   Theses authors contributed equally to this work.

**Abstract:** Developing circular value chains for continuing the use of and reducing the waste of the resources of industrial processing would eliminate impairments to the environment. The generation of nutrient-dense byproducts and coproducts with high-moisture contents are considered to be an issue for global food industries. These byproducts and coproducts spontaneously undergo chemical, biochemical, or microbial deteriorations due to high storage-temperatures, and consequently are turned into direct animal feed sources or even just treated as waste with eutrophication activity. This review provides an overview of selected mass-produced botanical food byproducts and coproducts (BFBC) including soybean okara, wheat germ, banana, and spent coffee grounds, with respect to value-added sustainable processing via proper drying technologies being employed. This review includes the current production of the above-mentioned agricultural products, the nutritional aspects of them, and the sustainable utilization of their coproducts. Additionally, the possible drying kinetics for value-added prospects are discussed.

**Keywords:** soybean okara; wheat germ; banana; spent coffee ground; coproducts; drying; kinetics

---

## 1. Introduction

The linear economy, the logic of "take, make, and dispose", has dominated since the beginning of the third industrial revolution [1]. Without building circular value chains, the unsustainable waste of byproducts and coproducts due to the linear production model causes serious impairments to the environment [2]. The circular-economic paradigm aims at eliminating waste and advocates the continual use of resources, suggesting the improvement of the durability of equipment, facilities and infrastructure, as well as the recovery of resources for other industrial processes [1–3]. There are more than enough reasons for scientists to devote endeavors into agricultural food disciplines and to pursue sustainable processing in a timely manner. For instance, the global neighborhood has become short of nutrients, and even just food calories, partly because some agricultural food materials do not meet commercial standards and thus are classified into the category of remnants, and sometimes only due to their sizes and shapes [4]. For others, the unsustainable food byproducts and coproducts are also problematic with respect to environmental issues [5]. Most of the mass-produced food byproducts and coproducts appear to be simultaneously collected when the final product is produced, and are withheld at warm or room temperatures, containing high moisture content. Followed by spontaneous occurrences of chemical, biochemical, or microbial deteriorations, these "supposed-to-be" food materials are turned into direct animal feed sources or even just treated as wastes

with eutrophication activity [4,6–8]. The generation of nutrient-dense byproducts and coproducts with high organic content are reported to be an issue for global food industries, as the food by/coproducts are being poorly managed and somehow turned into waste, with eventual effects related to greenhouse gas emissions [9].

It is obvious that providing these botanical food byproducts and coproducts (BFBCs) to human beings rather than to livestock and poultry should be the better choice for environmental and social sustainability, as people fed on plant-based diets could become healthier at much lower environmental and social costs than on meat diets [10].

However, the moisture content of a BFBC is considerably high (greater than circa 80%). Low temperature storage is one of the best solutions regarding the quality control and shelf life issues of BFBCs; as this approach would prevent the occurrences of adverse impacts in terms of sustainable preservation due to the capability of immobilizing water molecules that leads to the reduction in water activity. Food processing facilities would operate on heavy use of refrigeration [2]. It is estimated that refrigeration systems use as much as 15% of the total energy consumed worldwide; in the US food industry, a more intensive electricity (25%) is used for cooling and refrigeration from around the end of last century [2]. Nevertheless, concerns might have arisen for the cooling of some BFBCs initially produced at high temperatures, such as typical cases of soybean okara and spent coffee grounds, where the bulk matrix volumes have somewhat disproportional or limited surface areas; therefore, the cooling efficiency would not be ideal, i.e., the hot spot at the center would undergo deterioration caused by chemical rancidity, enzymatic reactions, microbial fermentation, or their combined occurrences [11–13]. However, the prolonged maintenance endeavor of refrigeration still deserves serious consideration. The use of preservatives would be another alternative for extending the shelf life and keeping the qualities of BFBCs, albeit with suspiciousness toward the unnatural additives. The food related industry is one of the largest global industrial sectors, making it an important user of energy [14]. The threat of limited food security has been the global highlight since the last century where the attributes of economic growth, population increase, and climate change have impacted on the global food production system [5,15].

Drying processes, the use of heat as a driving force to vaporize water molecules, is one of the common practices in the global food industry. To reduce the moisture content in BFBCs by drying can be considered as an energy consuming processing. Significant amounts of energy are used in the conventional methods for drying wet BFBCs, because the latent heat for evaporating water demands intensive energy consumption [16]. There is no disagreement among food scientists that drying is an attractive approach where energy savings need to be addressed. It is also reported that the food and agriculture industries account for at least 10% of total energy consumption for all US manufacture industries [17–19]. However, there could be remedies for the energy efficiency. One is to rely on eco-friendly thermal sources such as solar energy. The other is to make the best use of efficient apparatuses such as microwave dryers and fluidized-bed dryers with assisted radio frequency or ultrasonic pretreatments [18].

Other interests might be raised concerning the value-added sustainable processes of corn, rice, rye, and other crop materials. Mass-produced soybean okara, wheat germ, banana pulp/peel, and spent coffee grounds are traditionally or literally considered as byproducts during the preparation of soymilk, nonwhole-wheat flour, banana jam, and brewed coffee. In the light of transforming these mass-produced "byproducts" into more promising "coproducts" prior to undergoing deterioration and thereafter being treated as food waste, we provide an overview of these selected mass-produced agricultural food coproducts including soybean okara, wheat germ, banana, and spent coffee grounds, with respect to value-added sustainable processing via drying. This review includes current production of the above-mentioned agricultural products, the nutritional aspects of them, and sustainable utilization of their coproducts. Additionally, the possible drying processes/kinetics for value-added prospects are discussed.

## 2. Production of the Selected Materials

According to the report of Food and Agriculture Organization (FAO) of the United Nations [20], the Americas produce over 82% of the world's soybeans, circa 250 million metric tons, and the production keeps increasing during 2013–2017. Asia and Europe are the second (circa 25 million metric tons) and third (circa 10 million metric tons) world producers of soybean, respectively.

The production of wheat is larger than any other commercial crop and wheat continues to be the most important food grain source for humans. Asia has a production quantity of at least 300-million metric tons and Europe and Americas have more than 200 and 100 million metric tons, respectively. Moreover, Africa and Oceania each have circa 40 million metric tons wheat production [20].

Banana is considered to be the earliest fruit cultivated by human. Nowadays, banana is the most produced and consumed fruit in the world. It is predominantly produced in Asia, the Americas, and Africa with the production of 60, 30, and 20 million metric tons during 2013–2017, respectively [20].

Coffee bean is a seed of the *Coffea* plant. The world's largest coffee producing region is Latin America with the production of 5 million metric tons. Coffee production in Asia is more than 2.5 million metric tons and continuously increases during 2013–2017. In Africa, the production is more than 1 million metric tons and has been relatively stable [20].

## 3. General Description of the Selected Materials and Their Coproducts

### 3.1. Soybean Okara

#### 3.1.1. The Plant

The scientific name for soybean is *Glycine max* (L.) *Merrill*. It is an annual herbaceous plant of *Leguminosae* and *Glycine*. Soybeans contain about 35% protein, 15% lipids, and 30% carbohydrates. The carbohydrate portion contains starch (1%), soluble sugars (10%), and dietary fiber. Among the soluble sugars are sucrose (5%), stachyose (4%) and raffinose (1%), and the dietary fiber portion contains pectin (30%), cellulose (20%) and hemicellulose (50%). Among soy protein, the content of essential amino acid is the highest in lysine, and methionine is the first limiting amino acid because of less content. However, the protein efficiency ratio used earlier to evaluate protein quality was tested in rats, and its methionine requirement equals to 1.5 times that of humans, so it did not really represent the protein efficiency ratio of soy protein in humans [21]. In addition to general ingredients, soybean's functional ingredients include phospholipids, tocopherols, plant sterols, flavonoids, and isoflavones, so it has a potential as a health product.

#### 3.1.2. Origin of Soybean Okara

Soybean okara is a byproduct used to manufacture processed soybean foods such as soymilk, tofu, and bean curd. Approximately 1.1 kg of soybean dregs is produced for every 1 kg of soybean milk or tofu [22]. The moisture content of fresh soybean dregs is about 75%–80%, while the crude protein in dry matter accounts for about 20%–25% and fat accounts for 8%–19%. Okara contains about 50% of dietary fiber, of which soluble dietary fiber accounts for 2%–6%. Oligosaccharides include e.g., raffinose (0.1%–0.9%), stachyose (1.4%–4.1%), etc. [23]; Polysaccharides (cellulose) accounts for the most (56% ± 0.9%), followed by hemicellulose (12.1% ± 1.2%) and lignin (11.7% ± 1.4%), and the others are pectin and starch. Soybean dregs also contain substances with health benefits such as isoflavones, saponins, lecithin. Due to the high dietary fiber content of soybean dregs, it can be excreted in combination with bile salts and bile acids, reducing the content of bile salts and bile acids in the body and accelerating the rate at which the liver converts cholesterol to bile. These effects indirectly lead to a decrease in the cholesterol content, which will reduce the incidence of cardiovascular disease [24].

### 3.1.3. Contained Nutrients in Soybean Okara

Phenolic compounds include many types such as simple phenolic organics, phenolic acids, flavonoids, and tannins. Phenolic compounds are currently known to have a variety of physiological activities like antioxidant, anti-mutagenic, antibacterial, and anticancer. It is also known to reduce the incidence of atherosclerosis and coronary heart disease. In general, the most important mechanism for achieving these effects comes from antioxidant activity and ability to scavenge free radicals. Vegetables, fruits, cereals, soybeans, etc. are the main crops that contain these chemopreventers. Soybeans, in addition to high-quality proteins, also contain other functional substances such as phenolic acids and isoflavones [25]. Phenolic acids account for 28%–72% of total phenolic compounds in soybeans [26]. Studies have analyzed 43 phenolic compounds in soybeans, showing that in addition to isoflavones, the composition of phenols in soybeans include ferulic 95 µg/g, benzoic acid 57 µg/g and protocatechuic 44 µg/g. Others include vanillic acid, homogentisic acid, p-coumaric acid, catechin and quercetin [27]. Another study indicates that the phenolic compounds in soybeans still contain syringic acid, gentisic acid, salicylic acid, etc. [28].

Soy protein amino acid composition is rich and uniform. Potters et al. [29] have shown that feeding soy protein to rabbits and rats can significantly increase their bile acid secretion, which in turn increases the rate of cholesterol synthesis by the liver and accelerates cholesterol excretion. The digested peptide may also bind to bile acid. Many studies by means of enzymes to hydrolyze soybean protein to produce biologically active peptides have shown antioxidant and blood pressure-lowering effects [30,31]. Antioxidant peptides are usually composed of three to six amino acids. The nitrogen terminus mainly contains hydrophobic amino acids such as Val or Leu, and the sequence usually contains Pro, His or Tyr [32]; Davalos et al. [33] also confirmed that amino acids such as Trp, Tyr, Cys, His, Pro, and Met have antioxidant properties. The imidazole ring in the His structure has the ability to chelate metal ions and scavenge free radicals [34]. The tri-peptide Pro–His–His has the most antioxidant capacity, and His has a stronger antioxidant power at the nitrogen terminus (N-terminus) than in the middle and carbon terminus (C-terminus) of the peptide [35,36]. However, if His–His is removed, its antioxidant capacity will be significantly reduced.

## 3.2. Wheat Germ

### 3.2.1. The Plant

Wheat is one of the first domesticated food crops and has been the basic staple food in Europe, West Asia and North Africa for 8000 years [37]. The wheat kernel has an elliptical shape and is composed of 83% endosperm, 2.5%–3.5% germ and 14.5% bran. The endosperm consists of the aleurone layer and the starchy endosperm which is covered by the aleurone layer. It is the large central portion of the kernel and contains most of the starch and protein. The bran is the hard-outer layers of wheat grain and is composed of non-digestible cellulose.

### 3.2.2. Origin of Wheat Germ

Wheat germ is located in the base of kernel with lower portion. Wheat germ is the byproduct of wheat milling process. The yield of wheat germ is circa 0.4–0.5% and normally is carried over with the bran [38]. The world generation of wheat germ is estimated to be circa 2.5 million tons, annually [39]. Generally, the production of wheat flour involves cleaning, tempering and conditioning, milling, and blending. The vibrating separator is used to remove contaminants such as sticks and stones and other course and fine materials. The tempering and conditioning are done before milling to ensure the moisture content is uniform throughout the grain. Moisture would prevent the breakage of bran during milling. The wheat is then ground and sifted into flour, wheat germ, and wheat bran [40]. However, the moisture content of raw wheat germ is about 13% which would cause the lipid oxidation by enzymes easily, leading to the deterioration during storage [41].

### 3.2.3. Contained Nutrients in Wheat Germ

The wheat germ contains valuable micro- and macro-nutrients including dietary fiber, protein, oil, minerals, enzyme, and vitamins [42,43]. The protein is concentrated in wheat germ and is rich in essential amino acids, especially lysine and leucine [44]. Thus, it could be an alternative source of proteins for humans. Moreover, wheat germ is also rich in oil which is composed of 79.86% unsaturated fatty acids, including mainly linoleic acid (C18:2), oleic acid (C18:1), and linolenic acid (C18:3); and 20.14% saturated fatty acids, including mostly palmitic acid (C16:0) and icosanoic acid (C20:0) [45,46]. The poly-unsaturated fatty acids in wheat germ could prevent the cardiovascular disease and reduce the level of low-density lipoproteins and total cholesterol. Wheat germ contains more vitamin B, vitamin E, minerals, and functional compounds (flavonoids and sterol) than other parts in wheat kernel [38,43].

### 3.3. Banana

#### 3.3.1. The Plant

Banana is an evergreen monocotyledon, a fruit of a perennial giant herbaceous plant. It belongs to the subtropical plant of *Musaceae* (*Musa*) [47]. It is among the world's major food corps, after rice, wheat, and maize.

Edible bananas are domesticated from wild species and have diploid, triploid, or tetraploid hybrids of the A and B genomes. Musa acuminate is the ancestor of the A genome, and the B genome is derived from *Musa balbisiana.* These two species are diploids with the genomes AA and BB [48].

Bananas consumed nowadays are derived from the crossbreeding of *M. acuminata* and *M. balbisiana.* All current cultivars of bananas are hybrids and polyploids of these two bananas. *M. acuminata* cultivars are commonly used as dessert bananas, while *M. balbisiana* cultivars and hybrids of the two are used as cooking bananas. These cultivars are much more delicious than the original wild bananas. They are comparable to adult fingers and do not have large, hard seeds [47]. Bananas account for about 15% of the world's total fruit production [49], making it the fourth largest crop in the world, behind rice, wheat and corn. Cavendish (AAA) is the most traded banana, accounting for half of the world's banana production [50,51].

#### 3.3.2. Origin of Banana Pulp/Peel

The production of banana jam could result in the residue of its fruit pulp and peel. Moreover, these edible portions could also become byproducts during the production of dried banana slices consumed as snacks. Fruit peel is the main BFBC of banana accounting for 40% of the total weight of the fruit [52]. It has been treated as problematic food waste due to the massive amounts of organic materials [4].

#### 3.3.3. Contained Nutrients in Banana Pulp/Peel

The nutritional value of banana pulp is slightly different due to varieties. Generally, the 100 g edible portion of banana contains about 89.0 calories, 1.1 g protein, 385.0 mg potassium, 30.0 mg magnesium, 8.0 mg calcium, 1.0 mg sodium, 0.40 mg iron, 11.7 mg vitamin C, 610 µg of nicotinic acid, 40 µg of thiamine (vitamin $B_1$), and 23.0 µg of folic acid [53]. Bananas contain a large amount of total phenolic compounds such as gallic acid, catechins, epicatechins, tannins, and anthocyanins. These compounds impart an astringent taste to immature bananas. According to a previous research, the total content of phenols in bananas is 7 mg/100 g (fresh weight, FW) [54]. The content of free phenolic compounds (solvent extractables) in banana pulp is 11.8 to 90.4 mg (gallic acid equivalent, GAE)/100 g FW [55]. Banana peel is rich in dietary fiber, protein, essential amino acids, polyunsaturated fatty acids and potassium [56]. Banana peel also contains up to 3 g/100 g phenolic compounds on dry weight basis (DW) as well as catechin of 160 mg GAE/100 g DW [57]. It has also been reported that banana peel also possesses abundant carotenoids such as α and β-carotenes and lutein [58].

*3.4. Spent Coffee Ground*

### 3.4.1. The Plant

Coffee enjoys the reputation of "black gold" and is currently the world's second-largest trading commodity after petroleum [59]. Varieties, processing methods and roasting conditions are the key factors affecting the nutritional value and bioactive components of coffee beans. Coffee beans are rich in carbohydrates and are important precursors of aroma substances. During the roasting process, sucrose and arabinogalactan are cracked to produce formic acid, acetic acid, ethanol, lactic acid, 5-hydroxymethyl-2-furfural, HMF and other flavoring substances. The water-soluble polysaccharides produced after roasting can retain volatile substances and give a dense texture to the brewed coffee. Maillard reaction is a chemical reaction between reducing sugars and free amino acids, which gives roasted coffee beans a special aroma and flavor. The protein in coffee beans degenerates and degrades during the roasting process to form melanoidins. A previous study has shown that melanoidins are able to deliver antioxidant, antibacterial activities to human body and to act as a probiotic to regulate gut microbiota [60]. Coffee beans contain 10% lipids, which make an important contribution to the aroma of coffee. Lipids are involved in the production of alkylpyrazines, furanones, and phenol. Another study shows that Arabica beans contain more carbohydrates, fats and organic acids than Robusta beans. These ingredients have a considerable contribution to the presentation of coffee flavor. Therefore, Arabica coffee beans have a huge advantage in the market, with a market share of nearly 80%, and are the main raw material for making fine coffee [61].

### 3.4.2. Origin of Spent Coffee Ground

During the processing of coffee fruit, the shell, pulp, pectin layer and glume of the fruit will be removed. Finally, coffee green beans are left for subsequent roasting. These processes will produce a large number of byproducts, such as coffee husk, coffee pulp, and spent coffee grounds. Due to the large number of byproducts, compared to proper pre-treatment, which can be preserved for subsequent development of biomass energy or as a substrate for extracting phytochemicals, most coffee byproducts are often used as feed additives, fertilizers and deodorants. Coffee byproducts sometimes succumb to the high level of water activity and rich nutritional ingredients and are fermented and spoiled. The ill-treated directly disposed byproducts are not only environmental pollutants, but a waste of rich bioactive ingredients.

### 3.4.3. Contained Nutrient in Spent Coffee Ground

Previous studies used coffee shells and coffee pulp as substrates to ferment carotenoids via *Rhodotorula mucilaginosa* and discussed their antibacterial and antioxidant activities [62]. Studies on coffee grounds in recent years have confirmed that they are rich in polysaccharides, proteins, phenolic compounds, melanoidins and dietary fiber, and have antibacterial, antioxidant, and anti-inflammatory effects [63]. Previous studies have also confirmed the potential of coffee byproducts as dietary fiber supplements and hydrophilic bioactive antioxidants [64].

## 4. Drying Processes for Mass-Produced Foods and Their Coproducts

*4.1. Soybean Okara Drying*

Soybean residue or okara is typically with an approximate 80% moisture content and other compositions, such as carbohydrate, protein, fat, and dietary fiber [65]. Dried okara is often adopted as a food material with much extended shelf life, which is beneficial to easier lightweight transportation and inactivate some anti-nutrients such as trypsin inhibitor and saponins [65–68]. Technologies involved in conventional air, microwave, vacuum-freezing, rotary, jet spouted bed or air jet impingement mechanisms alone have been used to study the drying qualities of okara [66,67,69–72]. On the other hand, some combined drying methods such as pneumatic tube with rotational drum, microwave with

vacuum and high-voltage electric field with conventional air have also been employed to improve resulted qualities of okara [65,70,73–75].

Li. et al. used different methods to dry the okara and summarized that the freeze-drying was found to be the best in water holding capacity, swelling capacity and lipid binding capacity [69]. Another study shows that okara undergone freeze-drying presented a severe protein denaturation; and the okara containing low moisture content was found to develop greater porous structure to give a high solubility [66].

Kinetics regarding drying qualities of food byproducts could be complicated while a combined method is applied. In the case of okara, the change in colors and the drying rate raised major concerns. Perussello et al. had also investigated the combined effects of pneumatic tube and rotational drum drying to produce dried okara powder with a final moisture content equal to or less than 3% [70]. The pneumatic tubing approach is considered as a pre-drying process operated at 130 °C, 150 °C and 170 °C to trigger the initial moisture content reduction from 74% to 64% with a slight change in color; however, such a pre-drying approach did not significantly promote the drying rate (shorten the drying time). With the additional consideration of drying rate, the rotational drum drying controlled at 50 °C, 60 °C and 70 °C was conducted and resulted in the shorter drying time than a combined method using the pre-drying of pneumatic tubing. They also reported that the longer the drying time, the greater the color change can be observed regardless the drying methods [70].

Microwave techniques had been used to study the kinetic changes of okara during the drying process. Indeed, microwaving can increase drying rates and provide smooth and rounded starches contained in okara [66]. Senguta et al. [76] found that the microwave-dried okara causes more nutrient losses than vacuum drying. Furthermore, the microwave drying presented obvious color change due to a high dry temperature during the process [66,76]. Conversely, combining microwave drying with vacuum drying can give better qualities than using microwave alone; the qualities can be similar to those resulted by freeze-drying, meanwhile decreased the drying time by more than 90% compared to the freeze-drying process [65,73].

Besides the aforementioned, air jet impingement drying and jet spouted-bed drying were also applied to study drying kinetics. The air jet impingement directly injects air flow onto the sample to form a thin layer. The air jet impingement drying had been examined with different air velocity (1.3 m/s, 1.8 m/s and 2.3 m/s) and inlet air temperature (50, 60 and 70 °C) using a response surface methodology design [72]. After the treatment, the moisture content, the drying rate and the related quality were investigated. The results suggest that an effective condition should perform at 70 °C with air velocity and loading capacity at 2.3 m/s and 3 kg/m$^2$, respectively [8]. The jet spouted bed drying is considered as a modified application of fluidized-bed dryer. Wachiraphansakul and Devahastin [71] studied the drying kinetics of okara using the jet spouted bed dryer. The experiment was divided into two ranges of inlet air temperature; lower inlet temperatures (55, 60 and 65 °C) and higher inlet temperatures (90, 110 and 130 °C) [71]. In their conclusion, the recommended parameters were found to be 55 °C, 0.65 m/s, and 25 cm for inlet air temperature (at the lower temperature end), superficial air velocity, and initial bed height, respectively; these recommended processing parameters used specific energy consumption about 1.63 MJ/kg evaporated water [71]. On the contrary, for the higher temperature end, it should be performed at 130 °C, 0.55 m/s and 20 cm and the specific energy consumption is reportedly about 2.84 MJ/kg evaporated water [71] or 130 °C, 1.5 m/s and 18 cm for inlet temperature, superficial air velocity, and initial bed height, respectively and the specific energy consumption is about 3.69 MJ/kg evaporated water [67].

High electric field (HEF) drying is also known as an electrohydrodynamic (EHD) method [74], which applies high electric field to reduce the drying time with the aid of hot air drying at 105 °C [74,75]. The results showed that the drying time was reduced from 15% to 40% compared to hot air drying [65,74,75]; the simulation results indicate that the drying kinetics fits Page's model [75] for both with and without applying HEF. In the meantime, the color of okara presented darker yellowish at the surface when applied the HEF [74]. In another numerical study, the MATLAB program (MathWorks

Inc., Natick, MA, USA) was used to analyze heat and mass transfer phenomena during okara drying by using a fixed bed dryer, which is modified from a spray dryer apparatus, and compared to experimental results [77]. The drying time and temperature parameters were used to control a system with relative humidity and air velocity [77]. They summarized that the heat and mass transfer simulation by MATLAB fits experiment data with a correlation coefficient or $R^2$ about 0.9944, when using finite different methods with spherical geometry representing okara. Moreover, the use of numerical methods can reduce the cost of experiment and give a faster result [77]. As the final drying product of soybean okara can be expected to be in powder or flour forms; air jet impingement, jet spouted-bed, and fluidized-bed drying would be preferably in the light of energy efficiency due to the increase of sample's surface area. Future studies should be conducted towards higher temperature applications, as previous studies have shown that higher temperatures would favor the promotion of anti-oxidant properties in soybean [78,79].

### 4.2. Wheat Germ Drying

Wheat germ is another mass-produced by product from wheat milling processing and containing many nutrients, such as carbohydrate, protein, fat, vitamins and minerals [39,80]. However, fresh wheat germ is not able to bear a long-term storage due to unsaturated fats and endogenous enzyme with high moisture content causing rancidity, which is produced by oxidation reaction [81]. Different drying processes are used to overcome this problem and the numerical method was also applied. Gili et al. studied the dehydration of wheat germ using a fluidized bed dryer with both mathematical modeling and experiment measurements [80]. In their experiment, the fluidized bed dryer was employed at different temperatures (90, 110, 130 and 150 °C) at constant air velocity (0.5 m/s) [80]. At 150 °C, the Becker model was not correctly described by the whole drying process [15]. Arrhenius equation was used to describe the water diffusion coefficient as a function of inlet air temperature by another group of food engineers; they reported the calculated activation energy of 39.27 kJ/mol [80].

In other studies, finite element simulation using the programming of COMSOL Multiphysics (COMSOL Inc., Los Altos, CA, USA), a general-purpose simulation software for modeling designs, devices, and processes in all fields of engineering, manufacturing, and scientific research, was employed to study drying process of wheat germ [43,79]. The wheat germ was dried with two different drying time (4 and 25 min) in a fluidized bed dryer at 80 °C [79]. After the experiment, the data were fitted using programmed mathematic modeling, which can be proven to be compatible with the experiment results [41,81]. In summation, the short drying time was significantly different compared to traditional method about 70% [16]. Furthermore, they suggested that the short heating time approach should control both cooling temperature and final moisture content at 40 °C and 7%, respectively. Conversely, the tradition approach should be practiced at 40 °C and 3.5% [81].

At the same year, Chan and Kuo studied the effect of time and temperature for drying wheat germ drying by using fluidized bed dryer [41]. The results showed that the reduction of moisture content depended on heating time and temperature. Meanwhile, the thermal input was reduced as a function of heating time and temperature. Thus, a high temperature with short processing time (120 °C and 3 min) was able to save more energy than the lower temperature and is suitable for the hot and humid climate in Asia, especially, during summer [41,81]. Drying of wheat germ at lower temperature (<80 °C) by using the fluidized bed dryer could be a preferable condition for preserving not only the flavor and color of wheat germ but also the processing energy as compared to drying at higher temperature (>90 °C).

### 4.3. Dried Pulp/Peel Flour of Banana (Fruit)

Banana has been long-known as a climacteric fruit with highly perishable nature. It is necessary to reduce moisture content of the possible edible parts to extend the shelf life of the counterparts, thus drying techniques have been applied on various banana coproducts with different maturity [82]. The health effects of the banana pulp and peel flour resulted from drying have also been studied [52].

Beside the aforementioned (Section 3) health benefits of banana due to its contents of antioxidant components (often referred to bioactive compounds), vitamins and nutrient precursors, as well as dietary fiber; it should be noted that resistant starch [81], one of the most important dietary fibers mainly obtained from green banana via drying, has increasingly gained attention for its capability of bringing about the production of 4C or 4C-less fatty acids for prevention of carcinogens in the human colorectal track [83,84]. Therefore, value-added sustainable drying processes can allow resistant starch RS derived from green banana to be one of the healthy food ingredients [85], promote the phenolic acid contents in green banana pulp flour (GBPuF) with improved physicochemical properties [85–88], and retain the color attributes of yellow/green banana peel flour (YBPeF/GBPeF) [89–91]. The unbalanced supply-demand cycle due to overproduction can also be reliably relieved.

By drying unripe peeled bananas (first stage of ripening) in a dryer tunnel with temperatures ranging 52 to 58 °C and air velocity ranging 0.6 to 1.4 m/s, the comparatively higher resistant starch content (58.5 g/100 g dry basis) can be achieved at 55 °C under 1.0–1.4 m/s [92]. A recent drying reported that treating GBPF with pulsed-fluidized bed agglomeration (95 °C, 0.3 m s$^{-1}$ pulse, 10 Hz, and 1 m/s air flow) resulted in a higher RS level with appealing physicochemical properties—a shorter wetting time and better dispersity in cold water [93]—whereas the combined treatment of ultrasonic and pulsed vacuum followed by convective drying does not reportedly promote the RS content and could even deplete such content under certain circumstances [83], and least favorable RS levels might possibly result from common oven drying (50–60 °C for 12–24 h) while GBPuF were being employed as samples [55].

GBPeF or YBPeF exert their beneficial effects by supplementing probiotic growth and providing bioactive compounds to human health [94,95]; nevertheless, proper drying techniques would prevent the adverse change of color attributes regardless enzymatic or non-enzymatic causes. Room temperature drying at 23 °C for days resulted in minimum or no evidence of Maillard reaction when for GBPeF, which presents color appeals comparative to freeze drying at −50 °C [52,96]. An exhaustive dehumidified drying (60 °C, 13 h) with the combination of freeze drying is reportedly to possess highest flour lightness while YBPeF sample peel being examined, however, retain lowest level of phenolic compounds [90].

Microwave provides expedited drying efficiency in terms of moisture migration towards ambient in any scientific sense and allows to retain most of bioactive compounds, total phenolics and antioxidant activity while treating YBPeF at 960 W for 6 min yet noted with reduce in lightness comparing to dehumidified drying (60 °C, 13 h) [89]. Regardless of the operation cost, freeze drying is versatile for maintenance of color against browning reaction for most of the banana coproduct parts, i.e., GBPeF, GBPuF, YBPuF, and YBPeF, with the greatest total of flavonoids reported in YBPeF [89]. Accordingly, banana flours from different sustainable parts collected at various ripening stages could potentially be utilized as wheat flour or other likewise flours, with less favorable, however, acceptable viscoelastic behavior for certain consumers with health awareness; studied products include gluten-free cakes, sponge cakes, bread, snacks, ice cream, and pasta [86,87,89,91,96–98]. As for considerations of color appealing attributes, drying processes facilitating moisture and oxygen depleting features would be preferred and can be studied in the future. On the other hand, high powered microwave (960 W in the case of Alkarkhi et al. [89]) drying would be a potential candidate while the energy conservation and phytochemical retention are priorities; however, the development toward industry production level and the investigation of possible thermal runaway due to microwave energy will deserve some extra endeavors.

## 4.4. Spent Coffee Ground Drying

Spent coffee or spent coffee ground is the residue obtained from the coffee extraction process. After extraction, 650 kg of the spent coffee remained from one ton of green coffee [99]. In the spent coffee ground, it contains protein, fiber and sugars, especially mannose and galactose [61,99–101]. The spent coffee also contained 55% to 80% of moisture content and it is necessary to be dried for extended shelf-life, less packing and storage costs, and easier transportation [102]. Moreover, dehydrated spent

coffee ground equally produces a calorific value of about 25 MJ/kg [99,100]. The spent coffee was dried by using a cyclonic, convective, or freeze dryer [102–104].

For a cyclonic drying study, the spent coffee ground was fed with different mass flow ratios of spent coffee ground and air flow with an inlet air temperature ranging from 59 to 271 °C. In such results, the particle size and density of dehydrated spent coffee grounds were affected by the moisture content and an increasing of the mass flow ratio and air temperature significantly increased drying rate and final moisture content of the dehydrated spent coffee ground [103].

In 2017, Martinez-Saez and team [104] investigated the use of spent coffee in biscuit. In their work, the spent coffee was dried with conventional drying and freeze-drying methods. The results showed that moisture loss at 40 °C with convection (oven drying) was significantly lower compared to temperatures at 70 and 100 °C with convection and the freeze-drying. The freeze-dried spent coffee was found to have the lowest water activity. However, the freeze-drying required the energy consumption of about 99 kW·h, which is more than the conventional drying method at 40, 70 and 100 °C (0.163 kW·h, 0.177 kW·h and 0.198 kW·h, respectively). Therefore, the most adequate condition for drying spent which they suggested should dry at 40 or 70 °C of the conventional method [104].

Another convective approach was used to dry the spent coffee ground by facilitating biofuels. This study divided experimental conditions into sixteen experiments by four different pressed sample thicknesses (5, 10, 15 and 20 mm) and air temperature (100, 150, 200 and 250 °C) with constant air velocity at 1 m/s. After performing the drying processes, the experimental data were fitted by mathematical models such as Lewis, Page and Modified Page, Herdenson and Pabis, and Two Term Gaussian [100]. It is concluded that the Two Term Gaussian model was found the best for fitting with their experiment in all conditions. The effective moisture diffusivity values were between $1.29 \times 10^{-9}$ and $28.8 \times 10^{-9}$ m$^2$/s. and the activation energy was between 12.29 kJ/mol and 16.87 kJ/mol [100]. In the lights of drying efficiency and energy conservation, a thicker shaped sample, moderate air velocity, and higher convective temperature (20 mm, 1 m/s, and 250 °C, respectively in their case) would likely be recommended as a favorable process because the color appeal is not the issue for spent coffee ground. Additionally, the bioactive components such as antioxidants have been proved to be considerably resistible to convective oven drying [102,105]. However, thermal effects on specific compound will still deserved for further investigation.

## 5. Conclusions

Increasing demands on sustainable processing and circular economic trajectories regarding mass-produced coproducts have peremptorily become charismatic trends among food industries, as all possible food processing steps will result in coproducts, which will potentially cause impacts on the environment. It is apt to find proper value-added approaches via drying scenarios toward selected mass-produced food materials—soybean okara, wheat germ, banana, and spent coffee grounds. In this work, we revealed the current global productions of the selected source crops. In the light for evoking value-added potentials, soybean okara, banana pulp/peel, and spent coffee grounds contain considerable amounts of dietary fiber, bioactive phenolic compounds, and peptides as food ingredients to be commercialized. Additionally, the fatty acid profile of wheat germ is worth emphasizing for prevention of cardiovascular disease in addition to the contained vitamins and minerals. Air jet and fluidized-bed drying apparatuses provide promising solutions for the production of soybean okara flour. Fluidized-bed processes are preferable for wheat germ drying regarding retention of better product flavors. High power microwave energy is a reliable thermal source for phytochemical retention of banana pulp/peel as color appeal is not critical to quality. Convective dryers supplemented with moderate air velocity are good candidates for the drying of spent coffee grounds. For color appeals, drying with low ambient temperature, humidity, and oxygen levels, such as with freeze-drying techniques, would be preferable for all the selected BFBCs as retaining bioactive compound levels is a major concern. Value-added drying processes for selected BFBCs are undisputedly considered to be confirmed through future studies with respect to their health benefits.

**Author Contributions:** Conceptualization, H.Y. and M.-I.K.; methodology, H.Y., T.S. and W.-Y.Y.; validation, H.Y. and M.-I.K.; investigation, H.Y., T.S., W.-Y.Y. and M.-I.K.; resources, H.Y. and M.-I.K.; data curation, H.Y., T.S. and W.-Y.Y.; writing—original draft preparation, H.Y. and T.S.; writing—review and editing, H.Y. and M.-I.K.; visualization, H.Y. and M.-I.K.; supervision, H.Y. and M.-I.K.; project administration, H.Y. and M.-I.K. All authors have read and agreed to the published version of the manuscript.

**Funding:** This research received no external funding.

**Conflicts of Interest:** The authors declare no conflict of interest.

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
