# Peer review of "Drying Applications during Value-Added Sustainable Processing for Selected Mass-Produced Food Coproducts"

_processes, doi:10.3390/pr8030307_

Round 1

Reviewer 1 Report

Presented article contains basic characteristics and drying process of selected some agricultural food materials (such as soybean okara, wheat germ, banana, and spent coffee ground) based on literature data. Generally, this review is well written and based on adequate and new literature data.  

I have two main comments which should be taken into consideration before publication of this paper.

After each chapter concerning the drying process of discussed raw materials a short summary (1-3 sentences should be included) should be included: Which condition of drying should be used based on literature date. What should be next step in drying of such products....., what pre-treatments...., not only dry information based on literature data. The chapter 5 and 6 should be deleted, because it concerns in the small extent of described raw materials. Some information concerning drying kinetics of described products was added previously, for example line 264-274, or 284-290, or 408-415. Moreover, there is no any information in the conclusion concerning drying kinetics. Therefore this subject can be a basis for separate publication.

Besides:

Line 89: Change the bracket direction

Line 267. Please start “the” from a capital letter.

Line 385 please add "and pasta" and include the citation “Biernacka et all., Banana Powder as an Additive to Common Wheat Pasta Foods 2020, 9, 53; doi:10.3390/foods9010053”.

Author Response

Dear Reviewer:

Thank you for giving us the opportunity to submit a revised draft of our manuscript titled “Drying applications during value-added sustainable processing for selected mass-produced food co-products” to Processes MDPI- Special Issue "Drying Kinetics and Quality Control in Food Processing". We appreciate the time and effort that you have dedicated to providing your valuable feedback on our manuscript. We are grateful to your insightful and experienced comments on our paper. We have been able to incorporate changes to reflect all of the suggestions provided and have highlighted the changes within the manuscript.

Reviewer 2 Report

The paper entitled "Drying applications during value-added sustainable processing for selected mass-produced food co-products" was planned by authors as a review to provide an overview of selected mass-produced agricultural food co-products. The subject of the manuscript sounds interesting for a potential reader, but unfortunately the content of the paper needs revision. Authors should seriously rewrite the paper to be considered for the future publications.

Major comments:

  1. There is no explanation why these 4 products have been chosen. It is needed to present clear justification. The reader could be interested in comparison between the same family products (as e.g. wheat, rye, corn, rice). Authors should avoid impressions that the choice seems random.
  2. The description of each products should be presented in a similar way and the same structure. It is hard to find the common aspects in the description of each product.
  3. The same situation as above can be noticed for the drying processes section. In this section only for spent coffee ground drying there is an information regarding the energy consumption (aspect that is in connection with the environment and energy mentioned at the beginning of the paper). Please unify the description of each considered product and describe the energy consumption, influance into environment etc..
  4. Section 5 and 6 should be rewritten as one section (explain the  difference between common and basic models as well). Please add detailed explanation of all 13 methods mentioned and discuss them in connection with 4 chosen products.
  5. The conclusion section is very poor. Please extend the section by a summary regarding comparison of analyzed products in connections with drying methods, mathematical models and utilization of co-products.

Minor comments:

1. The language should be slightly improved e.g l. 68 (global, not grobal), l. 69 (century, not centry), l. 77 (total, not toatl) etc.

Author Response

(The authors gave the same response as above.)

Round 2

Reviewer 1 Report

Manuscript was significantly improved.

Author Response

Manuscript numbered-721230

Special issue Reviewer, Processes MDPI

Minor revision

Dear Reviewer,

Thank you for giving us the opportunity to submit a revised draft of our manuscript titled “Drying applications during value-added sustainable processing for selected mass-produced food co-products” to Processes MDPI- Special Issue "Drying Kinetics and Quality Control in Food Processing". We appreciate the time and effort that you have dedicated to providing your valuable feedback on our manuscript. We are grateful to your insightful and experienced comments on our paper. We have been able to incorporate other minor grammar changes and highlighted in red within the manuscript.

Sincerely yours,

Meng-I Kou

Huaiwen Yang

Reviewer 2 Report

Dear Authors

The modifications introduced in the manuscript seems reasonable, unfortunatelly, the paper in the current shape is hard to be accepted for publication. Some minor additional corrections are needed based on following comments:

Minor comment:

Comment 1:

Conclusions should be rewritten. Although authors included some modifications, the current shape seems rather as the abstract, not conclusions. The sentence ‘The drying approaches facilitated with or without pre-treatments are analyzed and justified' does not deliver any conlusion. Please add the short summary regarding analyzed approaches in the global context and 4 analyzed product context. The same comment is valid for kinetic models, drying effects (next sentences). The conclusions starting from l. 473 (‘For color…’) looks acceptable.

Comment 2.

There is still problem with the language.

The text should be checked by e.g. a native speaker or professional translation office. The updated version consists of new errors as  e.g. l. 82 – ;meight’ instead of ‘might’, l. 82 – ‘suistanble’ instead of ‘suistanable’, l. 84 – ‘traditinally’ instead of ‘ traditionally’, l. 342 – ‘stduies’ instead of ‘studies’ etc. Also the text should be checked from gramatical point of view – l. 354 – ‘,the Becker model was not correctly described the whole drying proces…’  - is ‘by’ missing?, l. 355 – ‘Another results, the Arrhenius-type equation was used to describe…’ - rewrite, etc.

Summing up: Authors should provide certicate that the manuscript has been checked.

Comment 3.

Authors removed section 5 and 6, but still in the manuscript exist names of various models (e.g. Page, Becker, Lewis, Gaussian, and others). Please add corresponding references in the text

l. 439 – add reference for Martinez-Saez

l. 358 – add reference for ‘COMSOL Multiphysics’

Summing up: Authors should add references for each new term introduced in the text. The manuscript is a review, so the reader should have a clear statement about the source of the knowledge and technology used.

Comment 4.

a) The text is not unified regarding dash and hyphen. Please correct that issue in the entire manuscript.

b) unify the presentation of the power e.g. l. 392 - 's-1' (check the entire manuscript)

Author Response

Manuscript numbered-721230

Special issue Reviewer, Processes MDPI

Minor revision

Dear Reviewer,

Thank you for giving us the opportunity to submit a revised draft of our manuscript titled “Drying applications during value-added sustainable processing for selected mass-produced food co-products” to Processes MDPI- Special Issue "Drying Kinetics and Quality Control in Food Processing". We appreciate the time and effort that you have dedicated to providing your valuable feedback on our manuscript. We are grateful to your insightful and experienced comments on our paper. We have been able to incorporate changes to reflect all of the suggestions provided and have highlighted the changes in red within the manuscript.

Here is a point-by-point response to your comments and concerns 

Minor comment:

Comment 1:

Conclusions should be rewritten. Although authors included some modifications, the current shape seems rather as the abstract, not conclusions. The sentence ‘The drying approaches facilitated with or without pre-treatments are analyzed and justified' does not deliver any conclusions. Please add the short summary regarding analyzed approaches in the global context and 4 analyzed product context. The same comment is valid for kinetic models, drying effects (next sentences). The conclusions starting from l. 473 (‘For color…’) looks acceptable.

Response: As per the suggestions of yours, we have rewritten the conclusions section. The changes can be found in lines 470-483.

Comment 2.

There is still problem with the language.

The text should be checked by e.g. a native speaker or professional translation office.

Response: We have asked a colleague who is a native English speaker to go through the text for clarity and grammar.

The updated version consists of new errors as  e.g. l. 82 – ;meight’ instead of ‘might’, l. 82 – ‘suistanble’ instead of ‘suistanable’, l. 84 – ‘traditinally’ instead of ‘ traditionally’, l. 342 – ‘stduies’ instead of ‘studies’ etc. Also the text should be checked from gramatical point of view – l. 354 – ‘,the Becker model was not correctly described the whole drying proces…’  - is ‘by’ missing?, l. 355 – ‘Another results, the Arrhenius-type equation was used to describe…’ - rewrite, etc.

Summing up: Authors should provide certificate that the manuscript has been checked.

Response: We have asked a colleague who is a native English speaker to go through the text for clarity and grammar including the new errors listed. The corrections can be found in Line 82, 84, 105, 223, 343-344, 356-358, 421-427, and 458. These corrections are also highlighted in red throughout the revised manuscript.

Comment 3.

Authors removed section 5 and 6, but still in the manuscript exist names of various models (e.g. Page, Becker, Lewis, Gaussian, and others). Please add corresponding references in the text

  1. 439 – add reference for Martinez-Saez
  2. 358 – add reference for ‘COMSOL Multiphysics’

Summing up: Authors should add references for each new term introduced in the text. The manuscript is a review, so the reader should have a clear statement about the source of the knowledge and technology used.

Response: We have accordingly added the reference for Martinez-Saez and numbered as [104]. This change can be found in line 442, which is highlighted in red.

COMSOL Multiphysics is a common software for engineering simulation and a said description ” (COMSOL, Inc. Los Altos, CA, USA), a general-purpose simulation software for modeling designs, devices, and processes in all fields of engineering, manufacturing, and scientific research,” has been added for clarity. This change can be found in line 360-361 of the revised manuscript highlighted in red.

Comment 4.

  1. a) The text is not unified regarding dash and hyphen. Please correct that issue in the entire manuscript.
  2. b) unify the presentation of the power e.g. l. 392 - 's-1' (check the entire manuscript)

Response: a) We have accordingly unified the dash and hyphen throughout the entire manuscript. b) We have unify the unit presentations for clarity using m/s instead of m s-1 throughout the text. The changes can be found in line 395, 396, 398, 453, 456, and 459 of the revised manuscript highlighted in red.

Sincerely yours.

Meng-I Kou

Huaiwen Yang